# Microtensile Bond Strength of Fiber-Reinforced and Particulate Filler Composite to Coronal and Pulp Chamber Floor Dentin

**DOI:** 10.3390/ma14092400

**Published:** 2021-05-05

**Authors:** Anja Baraba, Samir Cimic, Matteo Basso, Andrei C. Ionescu, Eugenio Brambilla, Ivana Miletić

**Affiliations:** 1Department of Endodontics and Restorative Dentistry, School of Dental Medicine, Gunduliceva 5, 10 000 Zagreb, Croatia; miletic@sfzg.hr; 2Department of Removable Prosthodontics, School of Dental Medicine, Gunduliceva 5, 10 000 Zagreb, Croatia; scimic@sfzg.hr; 3Department of Dentistry, Galeazzi Institute, University of Milan, via R. Galeazzi 4, 20161 Milan, Italy; Matteo.Basso@unimi.it; 4Department of Biomedical, Surgical and Dental Sciences, Oral Microbiology and Biomaterials Laboratory, University of Milan, via Pascal, 36, 20133 Milan, Italy; andreiionescu_40@hotmail.com (A.C.I.); eugenio.brambilla@unimi.it (E.B.)

**Keywords:** microtensile bond strength test, fiber-reinforced composite, particulate filler composite

## Abstract

This ex vivo study aimed to compare the microtensile bond strength of fiber-reinforced and particulate filler composite to coronal and pulp chamber floor dentin using a self-etching adhesive system. Coronal dentin of 40 human molar teeth was exposed by cutting occlusal enamel with a low-speed saw. Teeth were then randomly divided into two groups (n = 20). The first group was left as is, while in the second group, pulp chamber floor dentin was exposed by trepanation. After placement of a self-etching adhesive system (G-aenial Bond, GC, Tokyo, Japan), groups were further divided into two sub-groups (n = 10) according to the type of composite: fiber-reinforced composite (EP, everX Posterior, GC, Tokyo, Japan) and particulate filler composite (GP, G-aenial Posterior, GC, Tokyo, Japan). Then, composite blocks were built up. Sticks (1.0 × 1.0 mm^2^) were obtained from each specimen by sectioning, then microtensile bond strength (μTBS) test was performed. Statistical analysis included one-way ANOVA test and Student’s *t*-test (*p* < 0.05). μTBS values were 22.91 ± 14.66 and 24.44 ± 13.72 MPa on coronal dentin, 14.00 ± 5.83 and 12.10 ± 8.89 MPa on pulp chamber floor dentin for EP and GP, respectively. Coronal dentin yielded significantly higher μTBS than pulp chamber floor dentin (*p* < 0.05), independently from the tested composites.

## 1. Introduction

Reduced coronal and radicular residual tissues due to caries, previous restorations, and tissue removal during access cavity and root canal preparation make the restoration of endodontically treated teeth challenging [1,2,3]. Pulp chamber floor features account for additional potential difficulty while restoring a tooth following the root canal treatment. According to SEM analysis by Kijsamanmith et al. [4], pulpal floor dentin showed irregular dome-shaped calcospherites of varying size and open dentinal tubules without smear layer since chamber floor is usually not contacted by cutting instruments during access cavity preparation. Tubule density averaged 24,500 tubules per mm^2^ [4] meaning that the tubule density of pulp chamber floor is lower than coronal dentin [5]. In addition to specific morphological characteristics, adhesion to pulp chamber dentin may also be affected by the use of irrigants, most often sodium hypochlorite and EDTA. Such treatments alter the dentin’s organic and mineral content and influence the interaction with the adhesive materials used for coronal sealing. These factors contribute to weakening the structure and composition of endodontically treated teeth [6], making restorations more prone to failure if placed on endodontically treated teeth than on vital teeth [7]. Resin composite materials are the first choice for direct restorations after endodontic treatment, including composites with short fibers as fillers which have improved mechanical properties compared to conventional composites based on particulate fillers only [8] and might thus help reduce restoration failures. The short fibers, when randomly oriented, provide isotropic reinforcing effect, meaning that the strength of the material is the same in all directions [9]. Bijelic-Donova et al. [10] showed that fiber-reinforced composite had significantly higher fracture resistance and higher compressive fatigue limits than particulate filler composite. Furthermore, glass fiber-reinforced composite substructure may aid in eliminating crack propagation and root fractures [11]. Another two studies showed that fiber-reinforced composite provided endodontically treated teeth with superior fracture resistance [12,13]. Among the fiber-reinforced materials that showed this positive behavior, EverX Posterior (GC, Tokyo, Japan) is a bulk-fill material placed in increment depths up to 5 mm, which simplifies and speeds up the placement of composite restorations, reducing technique sensitivity. The placement of larger increments of composite material and shorting the clinical procedure is tempting alternative especially when restoring endodontically treated teeth and according to scientific data the effectiveness of bulk composite materials is comparable to conventional resin composites [14,15].

However, to the best of the authors’ knowledge, there is a lack of data regarding the microtensile bond strength of short fiber bulk-fill composite to the pulp chamber floor compared to coronal dentin.

This study, therefore, aimed to compare microtensile bond strength of fiber-reinforced and particulate filler composite to coronal and pulp chamber floor dentin using a self-etching adhesive system.

## 2. Materials and Methods

### 2.1. Specimen Preparation

Forty sound human molar teeth, extracted for periodontal or orthodontic reasons, were obtained for the experiment under the approval of the Ethical Committee of the School of Dental Medicine, University of Zagreb, Croatia. After extraction, the teeth were thoroughly cleaned using brushes and curettes and stored in 1% chloramine solution at 4 °C until use. All teeth were randomly divided into two experimental groups (n = 20 per group, Figure 1). In the first group, coronal dentin was exposed by cutting occlusal enamel with a diamond blade mounted on a low-speed water-cooled saw (Isomet 1000, Buehler, Dusseldorf, Germany, running at 200 rpm) to obtain a flat dentin surface (Figure 2). The dentin surface was polished with sandpapers of increasing grit (400, 600, 1000) to form a smear layer on dentin’s bonding surface. In the second group, pulp chamber floor dentin was exposed by trepanation, after which chemo-mechanical instrumentation and root canal filling were performed. The root canals were instrumented using Reciproc instruments size R25 (VDW, Munich, Germany) according to the manufacturer’s instructions. During instrumentation, canals were irrigated with 2.5% NaOCl using a 27-gauge needle and a 2 mL syringe. Root canals were rinsed with 2 mL of 17% EDTA (pH = 7.7) for one minute to remove the smear layer, and final irrigation was conducted with saline. Root canals were dried using Reciproc paper points size R25 (VDW) and obturated with a Reciproc gutta-percha cone size R25 (VDW) and AH Plus sealer (DeTrey Dentsply, Konstanz, Germany). After obturation, excess gutta-percha was removed with hot pluggers 1 mm from the cementoenamel junction (Figure 2). All roots were stored at 37 °C for one week to allow for the sealer to set.

Both experimental groups were randomly divided into two subgroups (n = 10) according to the type of composite used for the restoration: fiber-reinforced composite (test, everX Posterior, GC, Tokyo, Japan) and particulate filler composite (control, G-aenial Posterior, GC, Tokyo, Japan) (Figure 1, Table 1).

The bonding surface in all four groups was washed with distilled water and gently dried with a dental unit air syringe (Kavo Primus, 1058 S/TM/C/G, Biberach/Riss, Germany) before performing the adhesive procedure. The bonding surface was prepared and bonded according to the manufacturers’ instructions using G-Bond (GC, Tokyo, Japan). After bonding, a composite resin block was built-up on the bonding surface, with the application of layers of the material not thicker than 2 mm for the particulate composite and not thicker than 5 mm for the bulk fiber-reinforced composite (Figure 2). Each layer was cured with a LED light (Bluephase, Ivoclar Vivadent, Schaan, Liechtenstein, 1200 mW/cm^2^, soft start) for 20 s, keeping the light tip perpendicular to the substrate and the tip 5 mm away from the dentin surface. In groups with the pulp chamber dentin exposed, resin block was built-up to the occlusal surface level keeping the light tip in contact with the occlusal surface. Where coronal dentin was exposed, a 5 mm high composite resin block was built-up. All specimens were then stored in distilled water at 37 °C for 24 h, and then they were embedded into acrylic resin (Orthocryl, Dentaurum, Ispringen, Germany) (Figure 2). Afterward, the teeth were longitudinally sectioned (Isomet 1000, same parameters as previously specified) to obtain 1 mm × 1 mm beam-shaped sticks (Figure 2). Before further testing, each beam was checked under a stereomicroscope (Olympus SZX-12, Optical Co, Europe, GMBH, Hamburg, Germany) to verify that the adhesive interface was perpendicular its long axis. Only beams with the latter characteristic were used in this experiment.

### 2.2. Microtensile Bond Strength Test

The microtensile bond strength was tested with a universal testing machine (Triax Digital 50, Controls, Milan, Italy). Each beam’s ends were glued with cyanoacrylate adhesive (Loctite gel, Henkel, Dusseldorf, Germany) to specifically designed metal plates. A tensile load was applied to each beam at a crosshead speed of 0.5 mm/min until it fractured (Figure 2 and Figure 3). The maximum load at failure was recorded in newtons (N). Beams were then observed under a stereomicroscope to assess the failure mode (adhesive, cohesive, or mixed). A failure at the dentin/adhesive interface was considered as an adhesive failure; if it occurred in the composite resin material or dentin, the failure was considered cohesive, and failure was considered mixed if it involved the dentin/adhesive interface and the composite resin material or dentin at the same time. The cross-sectional area at the fracture site was measured to the nearest 0.01 mm with a digital caliper (Roc International Industry Co., Ltd., Guangdong, China) so the bond strength at failure could be calculated and expressed in MPa.

### 2.3. Statistical Analysis

Statistical analyses were performed using JMP 10.0 software (SAS Institute, Cary, NC, USA). Normal distribution of data was checked using Shapiro-Wilk’s test, and homogeneity of variances was verified using Levene’s test. A two-way ANOVA model was used considering the dentin type and the resin composite as fixed factors. Subsequently, post hoc Student’s *t*-test was used to highlight significant differences between groups. In all tests, the level of significance was set at 0.05.

## 3. Results

Data were log-transformed to approach a normal distribution. Two-way ANOVA did not show a significant interaction between factors (Table 2), meaning that the tested types of composites had the same behavior on both tested dentin substrates. Coronal dentin showed significantly higher microtensile bond strength values than pulp chamber floor dentin (Table 3, Figure 4) irrespective of the tested composite type (*p* = 0.0079). There was no statistically significant difference in microtensile bond strength between test and control resin composites (*p* = 0.4617).

When visually inspecting the specimens after the test, no cohesive or mixed fractures were found. In both groups and sub-groups, failure types were adhesive between bonding and dentin. A post hoc power analysis was performed to assess the sample size needed to find statistically significant differences between materials, possibly to drive the planning of other subsequent research projects. Due to the intrinsic variability of responses of the microtensile test performed on dentine substrate, assuming an alpha value of 0.05 and a power conservatively set to 0.80, given the found SDs, to find a 20% difference between materials, it was calculated that some 275 specimens would be needed.

## 4. Discussion

The present ex vivo study showed that the microtensile bond strength values were significantly higher for coronal dentin than pulp chamber floor dentin for both tested composite resin materials. Thus, these results indicate that the bonding performance of a self-etch adhesive is affected by the type of substrate, e.g., coronal vs. pulp chamber floor dentin, and by potential chemical changes due to irrigants used during endodontic treatment.

In agreement with the current study results, previous studies have found bond strength to pulp chamber floor dentin to be lower than coronal dentin [4,16,17]. According to the literature, bond strength to dentine tissue is influenced by structural characteristics such as the diameter and number of dentin tubules as well as the relative amount of peritubular and intertubular dentin [18,19]. Pulpal dentin contains predentin, irregular secondary dentin, and a high tubule density with large diameters. All these variations contribute to making pulp chamber floor dentin a relatively challenging bonding surface [20]. Overall, bond strength of resin composites to dentin can be regarded as a summation of the individual bond strengths provided by surface adhesion, resin tags, and hybrid layer [21]. Taking into consideration the tubules density and diameter and the amount of peri- and intratubular dentin, hybrid layer formation is expected to be the crucial factor providing bond strength to coronal dentin, with little contribution from resin tags. However, in pulp floor dentin, resin tags would contribute most of the bond strength, while a reduced contribution by hybrid layer formation is due to the limited amount of intertubular dentin available [22]. Interestingly, Lohbauer et al. [23] showed that the formation of resin tags does not influence the bonding strength of a one-step self-etching adhesive such as the one used in the present study. This observation and the reduced hybrid layer formation when a self-etching adhesive is used for bonding to pulp chamber floor dentin may explain the low bond strength obtained on pulp chamber floor dentin in this study.

It is acknowledged that, due to the complex anatomy of the endodontic space, the mechanical instrumentation of root canals alone is not effective in removing microorganisms and debris from root canals [24,25]. As large areas of root canal dentin remain untouched by the endodontic instruments, using irrigant solutions is essential for lubrication, debridement, dissolution of microbial structures and biofilms, and removal of the smear layer prior to root canal obturation [26]. Sodium hypochlorite is routinely used during endodontic treatment. Such a principle was seen to induce the oxidation of some components in the dentin matrix, forming protein radicals that will compete with the increasing vinyl-free radicals created by the light-activation of resin adhesives to premature chain termination and incomplete polymerization [27]. The hypochlorite anion can also infiltrate mineralized collagen and destroy collagen fibrils [28]. Moreover, decreases in calcium and phosphorus levels and changes in dentin’s mechanical properties, such as elastic modulus, flexural strength, and microhardness, were reported after irrigation of root canals with sodium hypochlorite [29]. Therefore, it is likely that sodium hypochlorite solution, as used in the present study being considered gold standard for root canal irrigation, may negatively influence the bond strength to pulp chamber floor dentin, possibly explaining the lower bond strength results for the pulp chamber floor dentin. The experimental setup of this study was not designed to allow us to ascertain to what point the contribution of dentin characteristics rather than irrigant solution could have negatively influenced the bond strength to the pulp chamber floor. The latter question may be the primary aim of future studies. EDTA was used as a chelating amino acid in the current study to remove the smear layer before root canal obturation. Several studies have evaluated the effect of EDTA preconditioning on the bond strength of self-etch adhesives to dentin and reported that EDTA was effective in improving dentin bonding for all-in-one adhesives [30,31,32,33,34]. This was not the case in the present study. However, it must be noted that EDTA was in contact with the pulp chamber floor dentin for only one minute, while the substrate was exposed to sodium hypochlorite for a prolonged time during the entire process of root canals instrumentation. Therefore, the detrimental effect of sodium hypochlorite on bond strength might have been more pronounced.

The current study also aimed to test the microtensile bond strength of two different types of composite materials, e.g., fiber-reinforced and particulate filler composite. Particulate filler composites have been present on the market for decades, but a more recent restorative dentistry approach uses fiber-reinforced bulk-fill composite as a base in extended restorations. One of the clinical advantages of using such a material is placing it in increments up to 4–5 mm, while achieving, at the same time, the mechanical performances needed to ensure extended restorations longevity. Bulk-fill materials can avoid several drawbacks of classical techniques, usually used for particulate filler composite material, including placing the material in 2-mm thickness increments. The latter exposes the restoration to the incorporation of voids between layers, failure of bonding, and extended treatment time [35]. Furthermore, short E-glass fibers added to the organic matrix of everX Posterior can stop crack initiation and propagation, increase fracture toughness and decrease the modulus of elasticity to values more similar to dentin tissues [36,37,38,39,40]. All these features may be especially beneficial when restoring endodontically treated teeth.

An earlier study showed that the bond strength to dentin was dependent on the composition of the restorative material [41] due to mechanical properties [42] or surface free energy characteristics of composites [43]. Steiner et al. [44] ascertained that the strongest influence on bond strength was exerted by the resin composite type, followed by the adhesive system, while the choice of the curing intensity was not significant. Therefore, it is essential to collect data about the bond strength of all composite materials becoming available on the market. Combined with other scientific data collected through in vitro, ex vivo, and in vivo studies, these results may help when choosing a restorative material. Indeed, according to Cekic et al. [45], the presence of fibers in the material and their orientation may also influence bond strength values. However, in the present study, no difference in microtensile bond strength between the tested materials, fiber-reinforced and particulate filler composites, was found, which is in agreement with the results of Tsujimoto et al. [46].

One of the indications for using fiber-reinforced composite is endodontically treated teeth, and this study found bond strength to pulpal floor dentin significantly lower than coronal dentin. Minimum bond strength of 17 MPa to dentin and enamel was identified as necessary to ensure successful adhesion [47]. In the present study, bond strength to pulpal floor dentin was lower than the 17 MPa threshold in both tested materials. For the particulate composite resin, it means that the polymerization shrinkage is greater than the bond strength to tooth structures, possibly resulting in marginal gap formation and composite failure [48]. However, the tested fiber-reinforced composite has a very low polymerization shrinkage strain (0.17%) primarily due to the random orientation of its fibers, which minimizes shrinkage during and after curing [45]. Jung and Park showed that resin-based bulk fill composites showed good marginal adaptation, even better than that of flowable bulk-filled materials. They assumed that a lower level of polymerization shrinkage and polymerization shrinkage stress was mainly responsible for this finding because it could induce less polymerization shrinkage force at the margin [49]. These considerations suggest that the tested fiber-reinforced composite might be a better choice than a particulate filler composite to restore endodontically treated teeth.

The microtensile test was used in this study to test the bond strength to coronal and pulp chamber floor dentin due to several advantages, such as multiple specimens being obtained from a single tooth and the stress more uniformly distributed during loading across the interface, especially for untrimmed specimens as the ones prepared for the present study [50,51]. It must be noted that the measured bond strength values may be considered reliable in the case of adhesive failures only. For both tested materials, only the adhesive type of failure was observed, indicating that the actual interfacial bond strength to dentin was determined.

The current study results must be interpreted with caution due to their possible limitations. The bond between hard dental tissues and restoration and its resistance to fracture is complex and cannot be correlated with the microtensile bond strength test results in a simple way. In vitro tests such as the used microtensile bond strength test cannot directly predict the clinical behavior of materials. Nevertheless, it may provide particular insight into the adhesion performances of restorative materials. Further research is necessary to investigate the influence of different types of adhesive systems on bond strength to such a peculiar substrate as pulp chamber floor dentin. Furthermore, thermal and mechanical cycling of specimens before bond strength tests can be considered when designing clinically relevant in vitro or ex vivo models of accelerated aging.

## 5. Conclusions

The present study showed that microtensile bond strength to pulp chamber floor dentin is significantly lower than coronal dentin when a one-step, self-etch adhesive is used. The microtensile bond strength values were not influenced by the type of composite used for the build-up. In a translational sense, fiber-reinforced and particulate filler composite are equally good alternatives for clinicians to restore vital and nonvital teeth.

## Figures and Tables

**Figure 1 materials-14-02400-f001:**
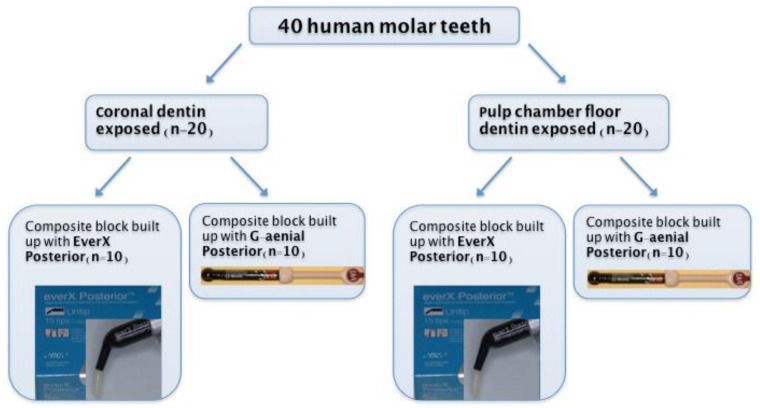
Experimental groups: 40 sound human molar teeth were divided into two groups according to the exposure of coronal dentin (n = 20) or pulp chamber floor dentin (n = 20). Each group was further divided into two subgroups depending on the material used for composite block build-up.

**Figure 2 materials-14-02400-f002:**
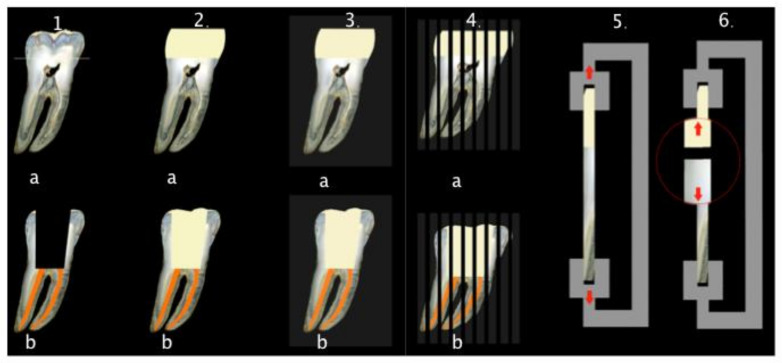
Experimental setup: (**1.a**) In the first group (n = 20), molar teeth had coronal dentin exposed by cutting of occlusal enamel using diamond blade of Isomet saw to obtain a flat dentin surface for adhesive procedure. (**1.b**) In the second group (n = 20), root canal treatment of molar teeth was performed. (**2.a**) and (**2.b**) In both experimental groups (n = 40), composite build-ups were made with the test (fiber-reinforced) or control (particulate filler) composite. (**3.a**) and (**3.b**) After composite build-ups, all teeth were embedded into acrylic resin in order to prepare them for sectioning. (**4.a**) and (**4.b**) Teeth were sectioned longitudinally using a diamond blade of Isomet saw to obtain 1 × 1 mm^2^ beam-shaped sticks. (**5.**) Each stick was glued to metal plates and placed in universal testing machine and a tensile load was applied at a crosshead speed of 0.5 mm/min until (**6.**) the stick fractured and maximum load at failure was recorded.

**Figure 3 materials-14-02400-f003:**
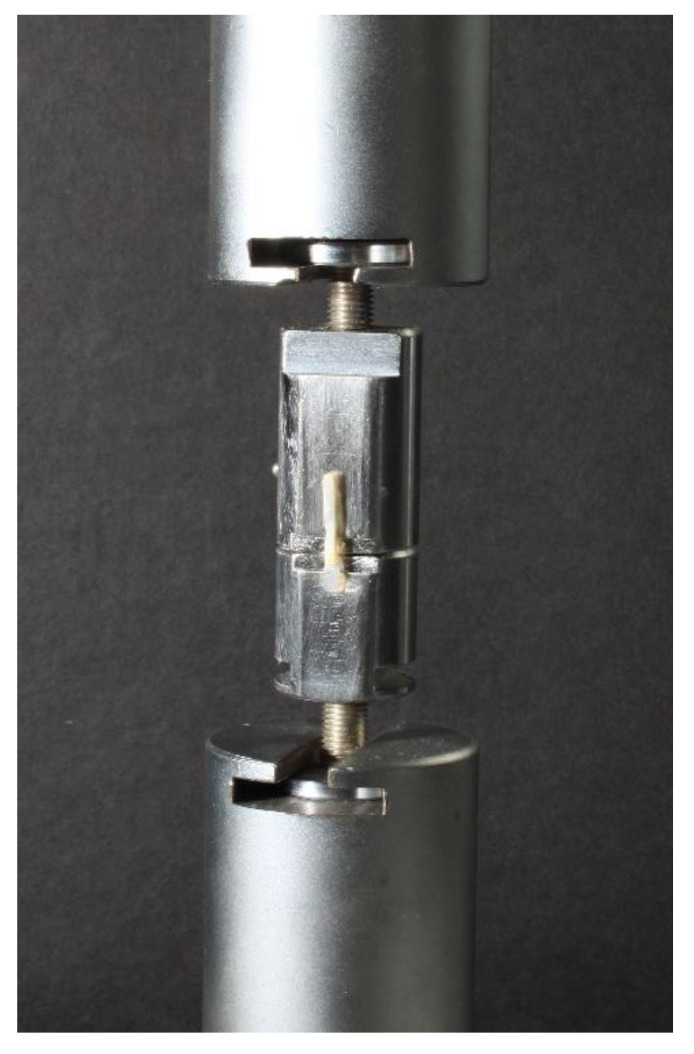
After obtaining 1 mm × 1 mm beam-shaped sticks by longitudinally sectioning teeth using Isomet saw, each stick was glued with cyanoacrylate adhesive to specifically designed metal plates and placed in universal testing machine Afterward, tensile load was applied to each beam at a crosshead speed of 0.5 mm/min until it fractured.

**Figure 4 materials-14-02400-f004:**
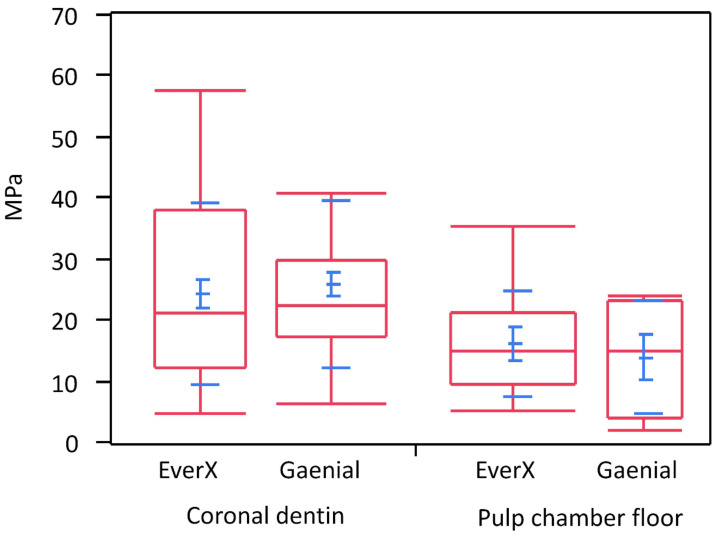
Box plot depicting the bond strength in MPa of the tested composite resin materials either bonded to coronal dentin or the dentin in proximity to the pulp chamber. The quartiles and 95% C.I. are depicted in red, while means +/− 1 standard error and +/− 1 standard deviation are shown in blue.

**Table 1 materials-14-02400-t001:** Chemical composition, weight percentage of the organic matrix, and volume percentage of fibers and fillers of the tested composites.

Material	Manufacturer	Composition
EverX Posterior (test)	GC, Tokyo, Japan	Bis-GMA, PMMA, TEGDMA, 74.2 wt%, 53.6 vol% short E-glass fibers, barium glass
G-aenial Posterior (control)	GC, Tokyo, Japan	UDMA, dimethacrylate-comonomers, 77 wt%, 65 vol% pre-polymerized silica/lanthanoid fluride fluoraluminosilicate/silica

**Table 2 materials-14-02400-t002:** Results of two-way ANOVA analyzing the effect of the two factors and their interaction. Star in superscript indicates a significant effect (*p* < 0.05).

Source	Nparm	Sum of Squares	F Ratio	Prob > F
Dentin location	1	1258.7711	6.9517	0.0098 *
Composite	1	0.3628	0.0020	0.9644
Dentin location*Composite	1	32.6946	0.1806	0.6719

**Table 3 materials-14-02400-t003:** Mean maximum load to failure of the microtensile bond strength test (MPa), standard deviation, and standard error for the tested groups and subgroups.

Level	Mean	Std Dev	Std Err Mean
Coronal, EverX	22.9089	14.6572	2.3777
Coronal, Gaenial	24.4409	13.7245	2.0019
Pulp, EverX	13.9967	5.8355	1.9452
Pulp, Gaenial	12.1040	8.8892	3.9754

## Data Availability

Data sharing is not applicable to this article.

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
