# Peer review of "Microtensile Bond Strength of Fiber-Reinforced and Particulate Filler Composite to Coronal and Pulp Chamber Floor Dentin"

_materials, 2021, doi:10.3390/ma14092400_

Round 1

Reviewer 1 Report

First of all congratulations to the authors for their work.

I appreciate that the paper is well written, following the rules of a scientific paper.

The subject is interesting for dentists, not only for specialists but also for dental practitioners. 

I appreciate very positively the Materials and Method section which demonstrates validity and reliability. 

I consider that the paper must be accepted in the actual form. 

Author Response

On behalf of all authors, I would like to thank the reviewers for the thorough review of the manuscript ‘Microtensile Bond Strength of Fiber Reinforced and Particulate Filler Composite to Coronal and Pulp Chamber Floor Dentin

Reviewer 2 Report

This manuscript has been already submitted and not sufficiently improved in the present version.

It deals with the in vitro investigation of microtensile bond strength of fiber reinforced compared with a particulate filler composite placed into two different dentin surfaces, coronal and pulp chamber floor. There are several major problems in all sections. Moreover, I suggest to the authors to read and carefully check the entire text of the manuscript, as there are several writing errors and oversights.

The introduction section does not sufficiently present the topic of fiber reinforced composites.  I suggest to re-write the entire section, explaining the characteristics of this material and his advantageous clinic applications. In addition, it is preferable to insert the sentences like "according to" (see lines 39-40, 60-64) in the discussion section. The paper must be updated. No recent references, highly relevant, are cited.

Material and Methods section is not well structured. Samples preparation is ambiguous, and the authors could explain in detail the Figure 2. Indeed, Figure 2 is not clear and does not correctly represent the step by step sequence of sample preparation, much less how they are evaluated and which instrument is used for their analysis

Regarding the Results section, Raw data should also be shown. The power samples analysis should be reported. In the table reported, some details are missing. I recommend writing the captions of the tables, as they must explain every particular content of the tables.

The Discussion shows some organization. Nonetheless, it should focus on the issues related to the composites used, discussing on the coronal and chamber bond strength. No relevant recent refs are mentioned.

The Conclusion of this manuscript in my opinion does not highlight a novelty for the dental clinician. The scientific impact as well as the clinical one are low. English language and style must be improved.

Again, I highly recommend to read more recent literature.

Author Response

On behalf of all authors, I would like to thank the reviewers for the thorough review of the manuscript ‘Microtensile Bond Strength of Fiber Reinforced and Particulate Filler Composite to Coronal and Pulp Chamber Floor Dentin. Suggestions were constructive and helped us to improve the quality of this manuscript. We agree with almost all their comments, and we have revised our manuscript accordingly. Please, find our point-to-point response.

Reviewer 3 Report

There are some weaknesses through the manuscript which need improvement. Therefore, the submitted manuscript cannot be accepted for publication in this form, but it has a chance of acceptance after a major revision. My comments and suggestions are as follows:

1- Abstract gives information on the main feature of the performed study, but some details about the background of the study (in a couple of sentences) must be added.

2- The introduction is too short and the literature study must be enriched. In this respect, authors must read and cite the following papers: (a) https://doi.org/10.1016/j.pdpdt.2018.12.014 (b) https://doi.org/10.1016/j.jmbbm.2019.02.009

3- Text in the figures (e.g., Fig. 1) is illegible.

4- As manuscript deals with experimental practice, authors must add some figure (in section 2.2) to show experimental test conditions.

5- In its language layer, the manuscript should be considered for English language editing. There are sentences which have to be rewritten.

6- The conclusion must be more than just a summary of the manuscript. List of references must be updated based on the proposed papers. Please provide all changes by red color in the revised version.

Author Response

(The authors gave the same response as above.)

Reviewer 4 Report

The article presents an interesting study. The study design is proper. However, manuscript has some major drawbacks.

Introduction

Please use other term for ‘short fiber composite’.

This section demands rewriting as is not based on recent references.

Materials and methods

Please add to this section 2 subsections, e.g. sample preparations, adhesive procedures.

Did study obtained Ethical Committee consent?

Figure 1 do not add any value to the article.

Subtitles of (all) figures are too long. Please write it in a less descriptive and vague way.

Results

Presents figure 3 as box and whisker plot.

Figure 3 what do you mean by ‘adhesive with dentine’?

Information on the failure type should be moved to the end of this section.

Discussion

This section demands rewriting as is not based on recent references.

How can you explain high standard deviation?

Lines 244-246 ‘An earlier study showed that the bond strength to dentin was 244 depended on the composition of the composite [37] due to mechanical properties [38] or surface free 245 energy characteristics of composites [39].’

Does adhesive system influence bond strength? Please note that the study (37) is almost 20 years old.

Lines 257-261 ‘If bond 257 strength values are less than 17 MPa, it means that the polymerization shrinkage of the particulate 258 composite resin is greater than the force adhering the material to the tooth which may result in the 259 marginal gap formation and composite failure [42].’

Reference 42 is a book published almost 20 years ago. Please find more recent and up to date articles, that will add value to the discussion.

References

Only 13 references were published in last 10 years. There most recent reference was published in 2017. Bibliography must be updated, most of the articles should be published within last 5, maximum 10 years.

Author Response

(The authors gave the same response as above.)

Reviewer 5 Report

The purpose of this ex vivo study was to investigate the microtensile bond strength of fiber reinforced and particulate filler composite to coronal and pulp chamber floor dentin. The design of the study is appropriate and the research quest to be addressed, to improve the knowledge about specificities of adhesion in endodontically treated teeth. The references are well cited and the methods adequate to address the research question.

However, there are some concerns which need to be addressed to improve the quality of the manuscript:

  1. Introduction needs to present the rationale and the significance of this topic in a better way, emphasizing the importance of the prevention of coronal microleakage in the long-term outcome of endodontic treatment, supported by in vivo models http://dx.doi.org/10.1016/j.joen.2013.10.023 and clinical research https://doi.org/10.1016/j.joen.2019.11.009 ;
  2. The present study is an ex vivo study, not an in vitro study;
  3. P4L111, use superscript for mm2 ;
  4. P5L125, define the direction of incident of the light source and its distance from the top layer of the composite;
  5. P5L129, please replace multiply for a more appropriate word;
  6. P5L132 mm2 use superscript again and check in all document;
  7. Table 2- replace “dentin” by “dentin” location to make it clear;
  8. There are no statistically significant differences between both tested materials in the present study, but may this be a consequence of the small sample size? Authors may perform a post-hoc power analysis to assess the sample size needed to find statistically significant differences between both materials. This information could drive the planification of other similar studies, by subsequent research projects.

Author Response

(The authors gave the same response as above.)

Round 2

Reviewer 2 Report

In this version, the authors have edited the manuscript according to the comments and suggestions correctly. Now, the work is presentable and well structured. However,  Materials and Methods section missing some details. The images are still confusing. Fig. 3 does not best represent the methodology used for the analysis. Also I would improve the captions, still unclear. References have been updated appropriately, however, they must be correct in style and form in the bibliography section.

Author Response

(The authors gave the same response as above.)

Reviewer 3 Report

The paper has been improved and corresponding modifications have been conducted.

Author Response

On behalf of all authors, I would like to thank the reviewers for the thorough review of the manuscript ‘Microtensile Bond Strength of Fiber Reinforced and Particulate Filler Composite to Coronal and Pulp Chamber Floor Dentin. Suggestions were constructive and helped us to improve the quality of this manuscript. 

Reviewer 4 Report

Introduction

Please add information on bulk-fill composite resin choice when used to restore endodontically treated teeth. You might use following papers:

Arbildo-Vega et al. Clinical Effectiveness of Bulk-Fill and Conventional Resin Composite Restorations: Systematic Review and Meta-Analysis. POLYMERS 2020. 12; 8; 1786

Chesterman, J et al Bulk-fill resin-based composite restorative materials: a review. BRITISH DENTAL JOURNAL 2017; 222 ;5 ; 337-344

Materials and methods

Please shorten the subtitle of figure 1.

Results

Presents figure 3 as box and whisker plot as described i.e. in https://en.wikipedia.org/wiki/Box_plot

Author Response

(The authors gave the same response as above.)

Round 3

Reviewer 4 Report

Dear Authors,

Thank you for the introduction of the recommendations.
The article can be published in the present form.

This manuscript is a resubmission of an earlier submission. The following is a list of the peer review reports and author responses from that submission.

Round 1

Reviewer 1 Report

First of all congratulations for the work. The subject is of interest in the field of dentistry.

The Abstract and The Introduction respect all the requirements and are well written.

Regarding the Results, I suggest to to explain more. They seem to be presented too shortly, in my opinion.

The discussion part is well written, but I could not see the limitation of the study and the necessity of further research. Also, the conclusion need to be presented

more larger.

Overall, I consider that the paper should be published with some revisions.

Author Response

On behalf of all authors, I would like to thank the reviewer for the thorough review of the manuscript ‘Microtensile Bond Strength of Fiber Reinforced and Particulate Filler 
Composite to Coronal and Pulp Chamber Floor Dentin’. All suggestions are constructive and will help to improve the quality of this manuscript. We agree with almost all their comments and we have revised our manuscript accordingly. Please, find our response to the comments.

REVIEWER 1

COMMENT 1: Regarding the Results, I suggest to to explain more. They seem to be presented too shortly, in my opinion.

REPLY 1: The results were explained in more details and two tables were added as well.

Results

            Two-way ANOVA did not show the existence of a significant interaction between the two factors (Table 2), meaning that the tested types of composites had the same behavior with regards to the dentin characteristics. Coronal dentin showed significantly higher microtensile bond strength values in comparison to the pulp chamber floor dentin (Table 3, Figure 3) irrespective of the composite type tested (p=0.0079). On the contrary, there was no statistically significant difference in microtensile bond strength between everX Posterior and G-aenial Posterior (p=0.4617). When visually inspecting the specimens after the test, no cohesive or mixed fractures were found. In both groups and sub-groups, adhesive failures were found between the applied bonding and dentin.

Table 2. Two-way ANOVA, Effect Tests

Source

Nparm

Sum of Squares

F Ratio

Prob > F

Dentin

1

1258,7711

6,9517

0,0098*

Adhesive

1

0,3628

0,0020

0,9644

Dentin*Adhesive

1

32,6946

0,1806

0,6719

Table 3. Mean maximum load to failure of the microtensile bond strength test (MPa), standard deviation, and standard error for the tested groups and sub-groups.

Level

Mean

Std Dev

Std Err Mean

Coronal, EverX

22,9089

14,6572

2,3777

Coronal, Gaenial

24,4409

13,7245

2,0019

Pulp, EverX

13,9967

5,8355

1,9452

Pulp, Gaenial

12,1040

8,8892

3,9754

(lines 197-192).

COMMENT 2: The discussion part is well written, but I could not see the limitation of the study and the necessity of further research.

REPLAY 2: Limitations of the study and the necessity of further research were included in the discussion. 

Nonetheless, the results of the current study must be interpreted with caution due to possible limitations of the study. Bond between hard dental tissues and restoration and its resistance to fracture is complex and cannot be simply correlated with the results of microtensile bond strength test. In vitrotests, like microtensile bond strength test used in the present study, cannot predict clinical behaviour of materials but may provide certain insight into the adhesion of restorative materials. Further research is necessary to investigate influence of different adhesive systems on microtensile bond strength of composite materials. Thermal and mechanical cycling of specimens prepared for bond strength tests could be beneficial for obtaining more clinically relevant results.  (lines 282-289)

COMMENT 3: Also, the conclusion need to be presented more larger.

REPLY 3: The conclusaion was presented more longer as suggested.

Conclusions

This study showed that microtensile bond strength values to pulp chamber floor dentin are significantly lower in comparison to coronal dentin when a self-etch adhesive is used.  The microtensile bond strength values were not influenced by the type of composite used for the build-up, fiber-reinforced and particulate filler composite seem equally good alternatives. The type of adhesive that was used, as in many biomechanical systems, may be considered the “weak point” when tensile forces are applied. 

(lines 290-296).

Reviewer 2 Report

Article is not appropriate to be published on materials.
Comment: The authors must reinforce all the experimental part which is very lacking. The idea is good but the experimental part is too poor to be published.

Author Response

We appreciate the Reviewer's opinion, however, at this moment, the authors are not planning to repeat the same experiment.

Reviewer 3 Report

In this study Barba and coworkers designed and performed an in vitro study to compare micro tensile bond strength of two type of composites of fiber reinforced and particulate filler to coronal and pulp chamber floor dentin using self-etching adhesive system. While the manuscript has a very limited data set, essentially only the micro tensile bond strength nevertheless it still contribute to the body of the literature and in this reviwer’s opinion can be published after revision.

  1. In the abstract, lines 26-27 and 29-30 there are two sentences which are saying the same thing please revise and remove one.
  2. The authors are advised to add several images from their work to make the MS an easier read for non-dentist reader of the paper. For example, images of coronal and pulp chamber floor dentine, a graphical abstract (GA) on the preparation of the specimens and application of the composites, image or GA for preparation of specimens/beams for micro tensile testing, images of the composites (fiber and particulate filler), etc. this will make the work mor interesting for the readers.
  3. Lines 45-58 indicates to the superior mechanical properties of short fiber composites compared to particulate filler, that being said the authors didn’t well justify the rationale of their objective, what has not been done in terms of testing and comparison of these two composites that make this work still interesting for the community? Micro tensile testing? If so why and what is the value of doing this? The introduction therefore needs to be revised.
  4. The composition of one of the resins, EverX has presented in line 48-50 while the composition of the second resin (G-aenial) is not clear/not mentioned. I suggest the authors to prepare a table clearly showing the composition and percent of each compound for both resins.
  5. Section 2.2. scanning electron micrographs of the samples to evaluate the failure after testing can be very useful
  6. The stat analysis was done using one-way ANOVA to compare the bond strength of the two composites while the authors could probably consider two way or GLM to compare both the composites and the coronal with pulp with regard to the type of composite. This could probably give more interesting information using the same dataset. As it is not currently clear if any of the composites could perform better depending on the location it has applied. Lastly the conclusion compared coronal to pulp “Higher microtensile bond strength values were observed for coronal dentin than for pulp chamber floor dentin” while this is not evaluated by the stat model.
  7. The discussion and conclusion need to be revised. For discussion more related articles can be discussed with the presented data.

Author Response

On behalf of all authors, I would like to thank the reviewer for the thorough review of the manuscript ‘Microtensile Bond Strength of Fiber Reinforced and Particulate Filler 
Composite to Coronal and Pulp Chamber Floor Dentin’. All suggestions are constructive and will help to improve the quality of this manuscript. We agree with almost all their comments and we have revised our manuscript accordingly. Please, find our response to the comments.

REVIEWER 2

COMMENT 1:In the abstract, lines 26-27 and 29-30 there are two sentences which are saying the same thing please revise and remove one.

REPLY 1: This part of abstract was slightly changed in order not to repeat similar statements but still to have a results and conclusion section in abstract.

Microtensile bond strength values were 22.91±14.66 and 24.44 ±13.72 MPa for coronal dentin, 14.00±5.83 and 12.10± 8.89 MPa for pulp chamber floor dentin for everX Posterior and G-aenial Posterior, respectively (p<0.05). Higher microtensile bond strength values were observed for coronal dentin than for pulp chamber floor dentin. (lines 27-30)

COMMENT 2: The authors are advised to add several images from their work to make the MS an easier read for non-dentist reader of the paper. For example, images of coronal and pulp chamber floor dentine, a graphical abstract (GA) on the preparation of the specimens and application of the composites, image or GA for preparation of specimens/beams for micro tensile testing, images of the composites (fiber and particulate filler), etc. this will make the work mor interesting for the readers.

REPLY 2:  Materials and methods section was modified by adding photos of composite materials used in the study in Figure 1. and by adding Figure 2. as a graphical presentation of the prepration of specimens for testing.

Figure 1. Experimental groups.

Figure 2.Figure 2. Experimental setup: 1.a) in the first group (n=20), coronal dentin was exposed by cutting of occlusal enamel with a diamond blade of the Isomet 1000 saw to obtain a flat dentin surface; 1.b) in the second group (n=20), root canal treatment was performed; 2.a) and b) both groups were further subdivided into two subgroups (n=10) according to the type of composite: fiber reinforced composite (everX Posterior, GC, Tokyo, Japan) and particulate filler composite (G-aenial Posterior, GC, Tokyo, Japan) which were placed in 5 mm high composite block in the group with exposed coronal dentin and up to the level of the occlusal surface in group with pulp chamber dentin exposed; 3.a) and b) the teeth were embedded into acrylic resin; 4.a) and b) teeth were multiply cross and longitudinally sectioned with a diamond blade of an Isomet 1000 precision, to obtain beam shaped sticks, with a cross-sectional area of 1mm2; 5. each stick was mounted in the universal testing machine and a tensile load was applied at a crosshead speed of 0.5mm/min; 6. the load was applied until it the stick fractured.

COMMENT 3: Lines 45-58 indicates to the superior mechanical properties of short fiber composites compared to particulate filler, that being said the authors didn’t well justify the rationale of their objective, what has not been done in terms of testing and comparison of these two composites that make this work still interesting for the community? Micro tensile testing? If so why and what is the value of doing this? The introduction therefore needs to be revised.

REPLY 3: To the best of the authors knowledge, there is lack of date on bond strength of fiber reinforced composite to pulp chamber floor dentin . Since one of the indications for the use of everX Posterior are endodontically treated teeth, it is beneficial to investigate its bond strength to pulp chamber floor dentin and compare it to bond strength to coronal dentin. This explanation is now included in the introduction section.

To the best of the authors knowledge, there is a lack of data regarding the microtensile bond strength of short fiber composite to pulp chamber floor in comparison to coronal dentin.  (lines 78-79)

COMMENT 4: The composition of one of the resins, EverX has presented in line 48-50 while the composition of the second resin (G-aenial) is not clear/not mentioned. I suggest the authors to prepare a table clearly showing the composition and percent of each compound for both resins.

REPLY 4: Table 1.  was added in the Materials and Methods section showing composition of both tested composite materials.

Table 1.Composition of EverX Posterior and G-aenial Posterior

Material

Manufacturer

Composition

EverX Posterior

GC, Tokyo, Japan

Bis-GMA, PMMA, TEGDMA,

74.2 wt%, 53.6 vol% short E-glass fibers, barium glass

G-aenial Posterior

GC, Tokyo, Japan

UDMA, dimethacrylate-comonomers,

77 wt%, 65 vol% pre-polymerized silica/lanthanoid fluride fluoraluminosilicate/silica

COMMENT 5: Section 2.2. scanning electron micrographs of the samples to evaluate the failure after testing can be very useful

REPLY 5: In the current study, SEM analysis was not performed, stereomicroscope was used to verify the failure mode. 

COMMENT 6: The stat analysis was done using one-way ANOVA to compare the bond strength of the two composites while the authors could probably consider two way or GLM to compare both the composites and the coronal with pulp with regard to the type of composite. This could probably give more interesting information using the same dataset. As it is not currently clear if any of the composites could perform better depending on the location it has applied. Lastly the conclusion compared coronal to pulp “Higher microtensile bond strength values were observed for coronal dentin than for pulp chamber floor dentin” while this is not evaluated by the stat model.

REPLY 6: We agree with the Reviewer, the statistical analysis was reperformed. After verification of normality of distribution and homoscedasticity, a two way ANOVA model was applied considering the dentin level and the composite type as fixed factors. Student's post-hoc t-test was used to highlight significant differences between groups at a significance level of 0.05. The results yielded by such analysis are indeed interesting and have been presented in the manuscript.

2.3. Statistical Analysis

            Statistical analyses were performed using JMP 10.0 software (SAS Institute, Cary, NC, USA). Normal distribution of data was checked using Shapiro-Wilk’s test and homogeneity of variances was verified using Levene’s test. Means and standard errors were calculated from the raw data. A two-way ANOVA model was used considering the dentin depth and the resin composite type as fixed factors. Subsequently, post-hoc Student’s t-test was used to highlight significant differences between groupsat a level of significance set at 5%.

(lines 160-166).

COMMENT 7: The discussion and conclusion need to be revised. For discussion more related articles can be discussed with the presented data.

REPLY 7: The conclusion was revised. Regarding the discussion section, according to the available and related data, the authors believe that all the relevant data were discussed.

Conclusions

This study showed that microtensile bond strength values to pulp chamber floor dentin are significantly lower in comparison to coronal dentin when a self-etch adhesive is used.  The microtensile bond strength values were not influenced by the type of composite used for the build-up, fiber-reinforced and particulate filler composite seem equally good alternatives. The type of adhesive that was used, as in many biomechanical systems, may be considered the “weak point” when tensile forces are applied. 

(lines 290-296).

Reviewer 4 Report

This manuscript deals with the in vitro investigation of microtensile bond strength of fiber reinforced compared with a particulate filler composite placed into two different dentin surfaces, coronal and pulp chamber floor. There are several major problems in all section.

The introduction section not presents the topic of fiber reinforced composite. Moreover, I suggest to explain the characteristics of this material and his advantageous clinic application. What about bulk-filling technique? Indeed, the authors claim that “the application of layers of the material not thicker than 2 mm for the particulate composite and not thicker than 5 mm for bulk fiber reinforced composite” (at lines 93-95), why this decision? Furthermore, no recent references, highly relevant, to this substantial contribution are given, also in Discussion section.

Material and Methods section is not well structured. Samples preparation is unclear, the authors could explain how different samples were made clarifying also using diagrams or figure.

Regarding the Results section, there is no statistical support of the findings. Raw data should also be shown. In the table reported are missing some details as statistical analysis.

In the Discussion section the authors focus mainly on problems derived by endodontic treatments, while the mechanical properties of fiber reinforced composite are not reported.

The Conclusion of this manuscript in my opinion not highlights a novelty for the dental clinician. English language and style must be improved.

Author Response

On behalf of all authors, I would like to thank the reviewer for the thorough review of the manuscript ‘Microtensile Bond Strength of Fiber Reinforced and Particulate Filler 
Composite to Coronal and Pulp Chamber Floor Dentin’. All suggestions are constructive and will help to improve the quality of this manuscript. We agree with almost all their comments and we have revised our manuscript accordingly. Please, find our response to the comments.

REVIEWER #

COMMENT  1: The introduction section not presents the topic of fiber reinforced composite. Moreover, I suggest to explain the characteristics of this material and his advantageous clinic application. What about bulk-filling technique? Indeed, the authors claim that “the application of layers of the material not thicker than 2 mm for the particulate composite and not thicker than 5 mm for bulk fiber reinforced composite” (at lines 93-95), why this decision? Furthermore, no recent references, highly relevant, to this substantial contribution are given, also in Discussion section.

REPLY 1: The topic of the current study is not fiber reinforced composite but the microtensile bond strenght of this material to pulp chamber floor and coronal dentin in comparison to particulate filler composite. Therefore, authors believe that fiber reinforced composite materials were described in sufficient details, keeping in mind the objective of the study. The clinical advantage of fiber reinforced material was mentioned in the introduction.

 According to Ozsevik et al.[9], placing fiber reinforced composite under composite increased the fracture strength of endodontically treated teeth to the level of intact tooth. EverX Posterior is a bulk-fill material which is placed in increment depths up to 5 mm which simplifies and speeds-up the placement of composite restorations and thereby reduces technique sensitivity. (lines…..)

COMMENT 2: Material and Methods section is not well structured. Samples preparation is unclear, the authors could explain how different samples were made clarifying also using diagrams or figure.

REPLY 2:  Materials and methods section was modified by adding photos of composite materials used in the study in Figure 1. and by adding Figure 2. as a graphical presentation of the prepration of specimens for testing.

Figure 1. Experimental groups.

Figure 2.Figure 2. Experimental setup: 1.a) in the first group (n=20), coronal dentin was exposed by cutting of occlusal enamel with a diamond blade of the Isomet 1000 saw to obtain a flat dentin surface; 1.b) in the second group (n=20), root canal treatment was performed; 2.a) and b) both groups were further subdivided into two subgroups (n=10) according to the type of composite: fiber reinforced composite (everX Posterior, GC, Tokyo, Japan) and particulate filler composite (G-aenial Posterior, GC, Tokyo, Japan) which were placed in 5 mm high composite block in the group with exposed coronal dentin and up to the level of the occlusal surface in group with pulp chamber dentin exposed; 3.a) and b) the teeth were embedded into acrylic resin; 4.a) and b) teeth were multiply cross and longitudinally sectioned with a diamond blade of an Isomet 1000 precision, to obtain beam shaped sticks, with a cross-sectional area of 1mm2; 5. each stick was mounted in the universal testing machine and a tensile load was applied at a crosshead speed of 0.5mm/min; 6. the load was applied until it the stick fractured.

COMMENT 3: Regarding the Results section, there is no statistical support of the findings. Raw data should also be shown. In the table reported are missing some details as statistical analysis.

REPLY 3:  We agree with the Reviewer, the statistical analysis was reperformed. After verification of normality of distribution and homoscedasticity, a two way ANOVA model was applied considering the dentin level and the composite type as fixed factors. Student's post-hoc t-test was used to highlight significant differences between groups at a significance level of 0.05. The results yielded by such analysis are indeed interesting and have been presented in the manuscript. A table reporting means, standard deviation and standard error for each group and subgroup was included.

Results

            Two-way ANOVA did not show the existence of a significant interaction between the two factors (Table 2), meaning that the tested types of composites had the same behavior with regards to the dentin characteristics. Coronal dentin showed significantly higher microtensile bond strength values in comparison to the pulp chamber floor dentin (Table 3, Figure 3) irrespective of the composite type tested (p=0.0079). On the contrary, there was no statistically significant difference in microtensile bond strength between everX Posterior and G-aenial Posterior (p=0.4617). When visually inspecting the specimens after the test, no cohesive or mixed fractures were found. In both groups and sub-groups, adhesive failures were found between the applied bonding and dentin.

Table 2. Two-way ANOVA, Effect Tests

Source

Nparm

Sum of Squares

F Ratio

Prob > F

Dentin

1

1258,7711

6,9517

0,0098*

Adhesive

1

0,3628

0,0020

0,9644

Dentin*Adhesive

1

32,6946

0,1806

0,6719

Table 3. Mean maximum load to failure of the microtensile bond strength test (MPa), standard deviation, and standard error for the tested groups and sub-groups.

Level

Mean

Std Dev

Std Err Mean

Coronal, EverX

22,9089

14,6572

2,3777

Coronal, Gaenial

24,4409

13,7245

2,0019

Pulp, EverX

13,9967

5,8355

1,9452

Pulp, Gaenial

12,1040

8,8892

3,9754

(lines 197-192).

COMMENT 4: In the Discussion section the authors focus mainly on problems derived by endodontic treatments, while the mechanical properties of fiber reinforced composite are not reported.

REPLY 4: The aim of the current study was to investigate the microtensile bond strength of two composite materials to pulp chamber floor and coronal dentin. Primarly, the bond strength will be influenced by the bonding supstrate (pulp chamber floor and coronal dentin). Pulp chamber floor dentin is very specific due to certain morphological characteristics and changes due to endodontic procedure, which could all influence the bond strenght of restorative materials. Therefore, mechanial properties of the fiber reinforced composite is not the main focus and, if disccused in greater details, might take the focus away from the objective of the study.

COMMENT 5: The Conclusion of this manuscript in my opinion not highlights a novelty for the dental clinician. English language and style must be improved.

REPLY 5: The conclusion was modified and English language was checked for spelling, grammar and language.

Conclusions

This study showed that microtensile bond strength values to pulp chamber floor dentin are significantly lower in comparison to coronal dentin when a self-etch adhesive is used.  The microtensile bond strength values were not influenced by the type of composite used for the build-up, fiber-reinforced and particulate filler composite seem equally good alternatives. The type of adhesive that was used, as in many biomechanical systems, may be considered the “weak point” when tensile forces are applied. 

(lines 290-296).

Round 2

Reviewer 3 Report

The authors responded to all my comments and in this reviewer's opinion, the paper is acceptable.

Reviewer 4 Report

I'm glad some of my advices have been taken into account. The effort that authors made to modify the manuscript is evident, but still not sufficient for the high standards required by the present journal. Indeed, the materials and methods and results sections are more clear and easier for the reader to understand now, but a statistical revision by an expert will be needed. In addition, it’s recommended to insert explanatory descriptive captions for each figure and table. Despite the changes made, there are still strong doubts about the scientific relevance that this study could bring. What would be the research novelty? Several scientific works underline the different microtensile bond strength between chamber and coronal dentin. It would be interesting to properly evaluate the behavior of two different composite materials in two different areas. In this sense, the authors should discuss their results and make a comparison, in order to strenghten their findings. Then, conclusions dealing with the adhesive topic is not properly addressed in the study. How can authors claim that the adhesive is the “weak point”? Have they studied different adhesive systems to conclude this? I suggest to extensively revise discussion and conclusion, perhaps as written in the previous version. Furthermore, it is not clear who is the potential reader of this article, whether researchers, dental clinicians or chemists. It should be explained.